# Evaluation of the design and implementation of municipal ice cleat distribution programs for the prevention of ice-related fall injuries among older adults in Sweden

**Robin Holmberg**[1,2]*, **Johanna Gustavsson**[1,2], **Carl Bonander**[2,3]

**1** Department of Political, Historical, Religious, and Cultural Studies, Karlstad University, Karlstad, Sweden, **2** Centre for Societal Risk Research, Karlstad University, Karlstad, Sweden, **3** School of Public Health & Community Medicine, Institute of Medicine, Sahlgrenska Academy, University of Gothenburg, Gothenburg, Sweden

* robin.holmberg@kau.se

**Data Availability Statement:** Data from the municipal survey are available at the OSF database (https://osf.io/vzd9u/). Regarding interview data:

## Abstract

### Introduction

The risk for outdoor falls tends to increase during winter due to icy road conditions. Several Swedish municipalities have introduced programs that provide their senior citizens with a pair of ice cleats in an attempt to tackle this problem. In this paper, we perform a process evaluation to identify potential barriers to the success of these programs and analyze the logic of their design.

### Methods

We sent a survey to all 290 Swedish municipalities to collect data on the characteristics of ice cleat distribution programs. We also performed focus-group interviews with older adults to gain insight into their thoughts about ice cleat programs. We synthesized our data with existing literature on ice cleats and behavior change theory to populate a logic model to identify and analyze hidden assumptions and potential flaws using program theory analysis.

### Results

On average, about 40% of the eligible population living in the intervention municipalities collected a pair of ice cleats. While we identified some other, but mostly minor, barriers to implementation, the main barrier appears to be a lack of scale (i.e., insufficient procurement and distribution of ice cleats), as 90% of all purchased ice cleats were eventually distributed. While previous research suggests that ice cleats can decrease injury risks if worn, we find that there is limited evidence on the effects of distribution on ice cleat use. Our interviewees emphasized the potential utility of ice cleats for staying safe and active during winter but stressed that ice cleats need to be user-friendly and of high-quality to increase the likelihood that a distribution program encourages behavior change.

The Swedish Ethical Review Authority has approved the study according to the Swedish Ethical Review Act (2003:460) regarding the ethical review of research involving humans. As dictated by the ethical body that approved the study and the promise to participants in their informed consent, the research data collected in the present study cannot be shared publicly as the data contain potentially identifying and sensitive personal data according to article 9 the General Data Protection Regulation (EU 2016/679), and public availability would compromise participant privacy. Data will be stored for at least 10 years at Karlstad University to enable review. Data are available for researchers who meet the criteria for access to confidential data. Data is covered by the Swedish Public Access to Information and Secrecy Act (2009:400) and a confidentiality assessment will be performed at each individual request. Permission from Karlstad University, Department of Political, Historical and Cultural Studies, has to be obtained before data can be accessed. Data requests may be sent to the Research data group at Karlstad University. Adress: Karlstad University, 651 88 Karlstad, Sweden. Tel. +46 54-700 10 00. E-mail: forskningsdata@kau.se.

**Funding:** This work was supported by a research grant from The Kamprad Family Foundation for Entrepreneurship, Research & Charity (grant number 20180067). The funders had no role in study design, data collection and analysis, decision to publish, or preparation of the manuscript.

**Competing interests:** The authors have declared that no competing interests exist.

## Conclusion

Existing ice cleat distribution programs appear to have reached a meaningful share of the targeted population. Additional research is required to assess their effects on ice cleat use and injury rates.

## Introduction

During winter, the northern hemisphere brings lowered temperatures, and some countries experience drastic weather changes as the temperature often drops down below the freezing point. With freezing temperatures comes an increased risk of slippery road conditions. In Sweden, two-thirds of all pedestrian falls that lead to serious injuries occur during the winter [1], and older adults are particularly likely to sustain an injury upon impact [2–4].

Traditional community-based interventions such as snow removal, road salting, etc., are frequently used to combat the seasonal rise in injuries in regions where snowfall is prevalent [5]. As a complement to these interventions and to promote active transportation [6], many Swedish municipalities provide a free pair of ice cleats to their senior citizens (aged 65+ years) to prevent ice-related fall injuries.

Gothenburg was one of the first municipalities in Sweden to implement an ice cleat distribution program. They partnered with retail stores around the city and distributed coupons to their senior citizens by mail. These coupons could then be exchanged for a free pair of ice cleats. Approximately 60% of the target population cashed in their coupons, and there is quasi-experimental evidence to suggest a short-term impact on ice-related fall injuries during the first year of implementation [7].

Since the introduction of the program in Gothenburg, many other Swedish municipalities have followed their example. However, knowledge about the design, quality of implementation, and impacts of the newer programs is limited. The utility of process evaluation and theoretical deconstruction of the logic behind a program have been emphasized in the program evaluation literature as a means to study the implementation of public health interventions [8–10]. The current paper aims to describe the implementation and characteristics of Swedish municipal ice cleat programs and analyze the logic of their design by applying a summative process evaluation.

## Methods

This study is a part of a project that aims to conduct a comprehensive evaluation of Swedish ice cleat distribution programs by assessing their implementation, impact, and cost-effectiveness. A pre-requisite for a causal impact of any intervention is successful implementation. Evaluations that investigate the quality of implementation are typically referred to as *process evaluations*, which can either be formative (aiming to improve ongoing programs) or summative (aiming to evaluate the quality of implementation without interfering with the program) [8]. The scope of the current paper is to investigate implementation and logic of ice cleat programs to provide a "qualitative flesh" to future impact evaluations, in which we will investigate their impacts on fall injury rates. To this end, we used a mixed-methods design [8–12] to conduct a broad-scale, summative process evaluation with a focus on program reach and the identification of other common barriers to implementation. To connect our data to potential program outcomes, we also critically analyzed the plausibility of a causal effect of ice cleat

distribution programs by constructing a program theory for Swedish municipal ice cleat programs, which is referred to as *program theory analysis* in the program evaluation literature [8,13].

This study is approved by the Regional Ethics Committee in Uppsala, Sweden. Approval number Dnr 2018/480. Written informed consent has been obtained from all participants in the study.

## Description of ice cleat programs and their implementation in Swedish municipalities

Swedish municipalities are local, self-governing authorities funded by municipal-level income taxes. They are responsible for providing several essential public services at the local level, including local traffic safety interventions. There are 290 municipalities in Sweden, with population sizes varying from 2,000 up to 975,000 according to population data from Statistics Sweden (2019). To investigate the characteristics and implementation of ice cleat programs in Sweden, we sent an electronic survey to all Swedish municipalities, asking them whether they ever had or have an ongoing intervention where they subsidize ice cleats to their citizens. If they did, we asked them to describe the program using a battery of questions related to implementation and program reach (see Results section for details), in addition to free-text responses that allowed the municipalities to elaborate on their answers. The response rate was 78.6% (n = 228). To verify that none of the non-responding municipalities had implemented a program, we searched their municipal websites, online local newspapers, and public decision documents but found no additional programs not covered by our survey data. In the results, we describe the implementation and the characteristics of a typical ice cleat program by tabulating the collected survey data. We also analyzed free-text responses to identify potential barriers for successful implementation.

## Analysis of program reach

To our knowledge, the municipalities did not set predetermined goals that can be used to define a successful ice cleat program. However, as a quantitative indicator of implementation success, we estimated program reach as the share of the target population who received a pair of ice cleats by combining data on the number of distributed ice cleats based on our survey with age-and-municipality-specific population data from Statistics Sweden. We also analyzed factors affecting reach by comparing groups of programs with above or below average reach. As an alternative measure of reach, we also considered the share of distributed ice cleats of the number of purchased ice cleats, which accounts for the fact that some programs were more limited in scale (i.e., not intended to reach the entire population).

## Program theory: Construction and analysis

We linked our data on implementation to potential causal effects on behaviors and health outcomes by constructing a program theory for ice cleat programs. Program theories can also serve as a tool for identifying hidden assumptions that may affect the likelihood of program success and the plausibility of the hypothesized outcomes given the activities conducted within the program [8,10]. Our program theory consists of a logic model that was populated based on empirical data and results from previous research. The model demonstrates hypothesized causal effects of ice cleat programs by illustrating the mechanisms required for successful implementation and how they relate to potential health outcomes [8,10] (an explanation of the model and its content will follow in the results section).

We used our survey data to construct an initial version of the logic model. In October 2020, we searched for previous research on ice cleats to supplement the model and analyze the plausibility of its assumptions, using combinations of the following keywords in PubMed and Scopus: ice cleat, anti-slip device, slip protection, anti-slip grip, anti-skid device, and prevention. We also snow-balled references from identified studies. It was beyond the scope of this study to conduct a formal systematic review. However, cross-referencing the results of our search with a systematic review of studies on pedestrian falls published in 2017 [4], which contains a section on ice cleats, yielded no new studies published before then. We also performed additional searches for published evidence and theory relevant for analyzing the underlying assumptions of the logic model, with keywords related to injury prevention, behavior change, and health promotion.

We also obtained supplementary data through focus group interviews to incorporate the perspective of the target population. The aim of the focus group interviews was to gain insight into why (or why not) the target population (older adults) use ice cleats as a fall prevention measure, their perception of ice cleat use, and what they think about a hypothetical ice cleat program. We recruited interview participants from a mid-western city in Sweden. Two independent local associations, with older adults as their target group, were contacted face-to-face and asked if they were interested in participating in focus group interviews about the use of ice cleats and municipal ice cleat programs. Participants were coordinated by the two associations independently and without any influence on participant selection from the researchers. In total, 13 senior citizens participated in the focus group interviews, a physically active gymnastics group ($n = 10$) and a less active Bridge group ($n = 3$). The gender distribution was nine women and four men, with an average age of 73. Prior to the interviews, we prepared a semi-structured interview guide with a battery of questions (see S1 Appendix) which was used as guidance during the sessions [14]. The interviews were conducted by one male and one female researcher (RH & JG). JG has a long background in health care, is a registered nurse, has a Ph. D. in risk and environmental studies, and is specialized in fall-preventive measures in the elderly. JG also has extensive experience in the field of qualitative interviews for research purposes. JG acted as the main interviewer and moderator of the sessions. RH is a Ph.D.-student and had limited prior experience in conducting interviews. Therefore, RH acted as a moderator assistant by managing the logistics, monitoring the recording equipment, and taking careful notes. One session was held at the gym and the other at the bridge club. No other external persons were present during the sessions besides the two researchers and the participants. Both sessions were recorded and lasted approximately one hour each, and no participants dropped out of the sessions. The collected data was transcribed and independently analyzed by two of the authors (RH, JG). We used a mix of both latent and manifest content analysis with an inductive approach to analyze and categorize the interview data [15,16].

We iteratively discussed the data and available evidence among the authors to reach a consensus about the construction and interpretation of the final logic model. The data and references in support of each central part of the logic model are described in detail in S1 Table. Below, we present the results of the program theory analysis in the form of a qualitative synthesis of our empirical data and data from previous research and theory identified in our search process.

## Results

### Characteristics and implementation of ice cleat programs

We begin with a general description of the characteristics of the Swedish ice cleat programs based on our survey data. A general overview of the survey data is provided in Table 1.

**Table 1. Descriptive statistics of municipal ice cleat programs for older adults based on a survey of 228 (out of 290) municipalities in Sweden.**

| | All programs (n = 73) | Below-average programs (reach <40.5%) (n = 29) | Programs above-average (reach >40.5%) (n = 44) |
|---|---|---|---|
| **Free or to a discounted price (in percent)** | | | |
| Free hand out | 84.9 | 86.2 | 84.1 |
| At a discounted price | 6.9 | 6.9 | 6.8 |
| "Free" in exchange for buying lunch/coffee | 8.2 | 6.9 | 9.1 |
| **Communication strategy (in percent)[α]** | | | |
| Traditional media | 60.3 | 62.1 | 59.1 |
| Social media | 46.6 | 51.7 | 43.2 |
| By mail | 16.5 | 13.8 | 18.2 |
| Personal communication | 30.1 | 37.9 | 25 |
| Other[β] | 21.9 | 31 | 15.9 |
| Average number of communication channels | 1.75 | 1.96 | 1.61 |
| **Distribution points (in percent)** | | | |
| Stores, pharmacies other businesses that provide ice cleats | 23.3 | 17.2 | 27.3 |
| Municipal owned buildings, like libraries or city halls | 45.2 | 58.6 | 36.4 |
| Pensioner organizations | 4.1 | 10.3 | - |
| Other locations[†] | 27.4 | 13.8 | 36.4 |
| **Type of ice cleats (in percent)** | | | |
| Whole foot devices | 72.9 | 70.4 | 74.4 |
| Heel devices | 2.9 | 7.4 | - |
| Forefoot devices | 2.8 | 3.6 | 2.3 |
| Unknown | 8.6 | 3.7 | 11.6 |
| Whole + Heel | 2.7 | - | 4.6 |
| User's choice | 11.4 | 18.5 | 7 |
| **Eligible age-groups (in percent)** | | | |
| 65 years and older | 84.9 | 89.7 | 81.8 |
| 70 years and older | 9.6 | 3.5 | 13.6 |
| 75 years and older | 4.1 | 6.9 | 2.3 |
| 80 years and older | 1.4 | - | 2.3 |
| Total number of eligible individuals | 492 388 | 219 942 | 272 446 |
| Total distributed pairs of ice cleats[‡] | 217 572 | 32 719 | 184 853 |
| Average distributed pairs of ice cleats | 3 296.5 (8 838.2*) | 1 128.2 (1 932.7*) | 4 996 (11 464*) |
| Share of purchased ice cleats that were distributed (in percent)[§] | 89.9 (19.9*) | 89.9 (23.8*) | 89.9 (17.1*) |
| Average program duration (in days) | 836 (791.5*) | 920.2 (904.3*) | 779.2 (711.2) |
| Average reach of the targeted population in the municipality (in percent) | 40.5 (24.6*) | 17.7 (11.7*) | 58.3 (15.5*) |
| Average cost per pair of ice cleats (in SEK) | 62.4 (45.9*) | 87 (56.9*) | 46.6 (28.4*) |

Notes: Survey data were collected between June to November 2019. Five municipalities were excluded as they distributed ice cleats to all citizens (all ages), which is beyond the scope of this study. Four municipalities distributed ice cleats before the region of Jönköping took over the distribution (the region consists a total of 13 municipalities), and their pre-region distribution data were excluded from the data in the table (reach to target population ranged between; 1.5% - 53%).

[α] = Percentages presented on these rows do not sum to 100% because the categories are not mutually exclusive.

[β] = Posters, brochures, senior organizations, meeting points for seniors, etc.

[†] = Fall-preventive weeks, senior meetings, health care centers, dedicated workshops, country stores, etc.

[‡] = Based on municipalities that reported the amount of distributed ice cleats (n = 66).

[§] = The relative reach is based on the municipalities that both reported their purchased amounts and distributed ice cleats (a total of 49 observations).

* = Std. Dev.

According to our survey, 78 out of 290 municipalities have implemented ice cleat distribution programs, 73 of which focused on older adults.

The average reach with respect to the eligible age group living in the program municipalities was 40.5 percent. The eligible population varied in age; the most common age group was those who have reached retirement age, i.e., citizens over the age of 65. Some municipalities targeted all citizens while others chose to distribute to people who are more fall-prone and older than others (usually citizens older than 75 of age who require home care, medical assistance, etc.).

The majority of municipalities offered a free pair of ice cleats, while a handful of municipalities offered ice cleats at a discounted price or in exchange for buying lunch/coffee at a municipally-owned restaurant. There are roughly three different types of devices applied on the sole of the shoe; they cover either the whole foot, the forefoot, or the heel of the shoe (for a detailed illustration, see reference [17]). Whole foot devices were the dominant type of ice cleats distributed by the program municipalities (72.9%).

The existence of the programs was typically communicated via multiple channels, the most common ones being traditional (e.g., newspapers) and social media (Table 1). Distribution occurred during the winter months (October to Mars). Municipal-owned buildings were the most common distribution sites, followed by other locations such as health care centers and stores that provide ice cleats.

**Differences between groups based on reach.** We divided the existing programs into two groups based on their reach to assess differences between the programs (above-average reach [n = 44, 60%] versus below-average reach [n = 29, 40%]). In general, the groups had similar characteristics. One eye-catching distinction is that the average costs per ice cleats from the below-average programs are almost twice as large as in the above-average programs. Also, the duration of the programs seems to be longer in the below-average municipalities (on average, almost five months longer). Distribution points also differ somewhat between the groups; distribution via retailers and "other locations" seem to be more frequent in the above-average programs (according to free-text responses, it appears that the "other" category in this group are mostly health care centers and country stores). However, the most striking difference between the two groups of programs is their scale; programs that reached a larger share of the population purchased many more ice cleats, which implies that they had the ambition or resources to cover a larger share of the population. Importantly, we note that data from both groups indicate that about 90 percent of all purchased ice cleats were eventually distributed (Table 1).

## Focus-group interviews

In the analysis of the interview data, three main categories were identified: *risk-aware and in control*, *ice cleats–a necessity* and *appreciates municipal ice cleat programs*. The content of the categories and illuminating quotes are presented in the following subsections.

In general, when exposed to winter conditions, the participants' primary choice to prevent outdoor falls was the use of ice cleats. Ice cleats facilitate an active life and are also seen as a simple yet inexpensive countermeasure to reduce slips and fall risks.

**Risk-aware and in control.** It was evident that there was a common concern about belonging to a vulnerable group and how falls could lead to serious injuries.

"*When you are younger, you have so many margins in life. If you are careless with one thing or another, you will not notice much. If you are careless now, then there can be consequences.*"

(Gym group).

At the same time, they felt confident that they could deal with the risks of falling.

"*We are getting wiser and wiser. You understand that you are past the age of 25. I mean, when I was 25, I could fall on the ice, and nothing happened. I just got up again. If I fall on the ice now, I will break myself!*"

(Gym group).

Their fear of falling would not lead them to live a sedentary lifestyle, because a sedentary life was considered harmful. Instead, they shared a strong desire to stay active, which could contribute to a healthier and longer life.

"*You need to stay informed about how important it is to be active and how dangerous it is to be sedentary.*"

(Bridge group).

**Ice cleats—A necessity.** Belonging to a vulnerable pedestrian group, together with the desire to uphold an active lifestyle, were strong motivational factors for ice cleat use. The occurrence of snow and ice would not hinder them from daily walks or running other errands. They do not have the time to wait for general countermeasures like snow removal, gravel, or road salting, which motivates the use of personal anti-slip equipment such as ice cleats.

"*I am so scared to fall! Because where I live, these big streets, spreading gravel is the very last thing they do.*"

(Bridge group).

The participants agreed that the use of ice cleats made it possible for them to cross icy roads and walkways, and they had the daily habit of assessing the environment before exposing themselves to ice and snow.

"*Of course, you look at the thermometer every morning and assess the risk before you pick up the newspaper.*"

(Gym group).

They noted that user-friendliness is a key determinant for continued use. However, they also had poor experiences with ice cleats that deteriorate quickly over time.

"*. . . but when you walk on gravel roads, the rubber gets worn down after three or four weeks. Then you have to go and buy new ones. . .*"

(Bridge group).

**Appreciates municipal ice cleat programs.** In general, the interviewees welcomed the idea of a municipal ice cleat program. They estimated that the municipal costs would be low and especially appreciated the signals that a program of this kind would send.

"*The municipalities do not have to hand out the best of ice cleats. The programs can instead create awareness;* "*Now we think of you and your well-being, take the chance!*""

(Gym group).

However, they saw some potential barriers. Free ice cleats would probably trigger people to pick up a pair, but they are probably going to be left unused. There were also reservations of a more ideological nature.

"*I also think it is a bit of a shame to take responsibility away from people just because they are a little older. . . I am hesitant about such programs.*"

(Bridge group).

The main hesitation to program introduction was the distributed ice cleats may not be user-friendly. They opposed programs that determine which ice cleats to use. A few participants also noted that they would prefer shoes with studs over ice cleats.

"*There are so many different types of ice cleats, so you want to choose the ones that suit you best.*"

(Gym group)

## Program theory analysis: Overview

We now turn to our program theory analysis, which centers on our proposed logic model for a typical ice cleat program in Sweden (Fig 1). The logic model provides a general description of a municipal ice cleat distribution program. The model is divided into two key parts: a process theory (including inputs, activities, and outputs) and an impact theory (including the expected causal effects of a successfully implemented program) [8]. The *inputs* are the resources that go

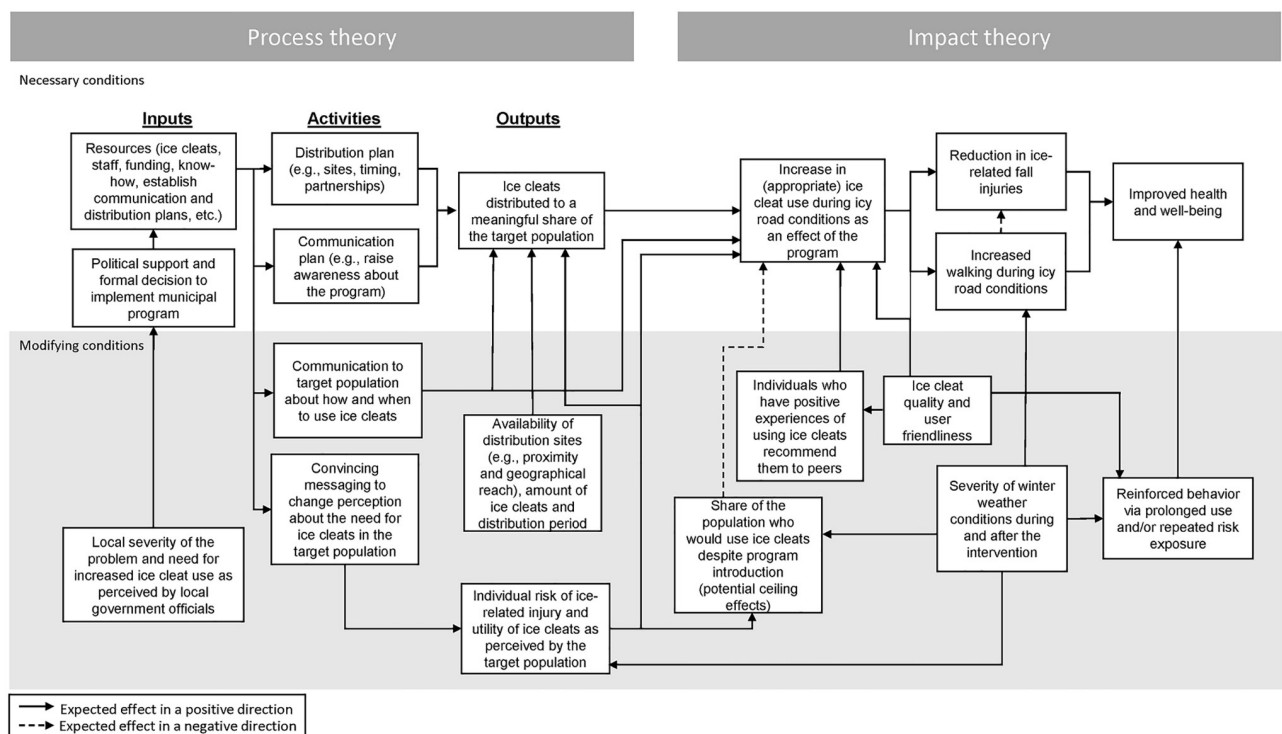

**Fig 1. Hypothesized necessary and modifying conditions for a successfully implemented municipal ice cleat program.**

into the program, such as political support, funding, staff, facilities, etc. The inputs are then translated into *activities* which services are provided by the program, such as communication campaigns, distribution plans, etc. Each activity is connected to various *outputs* provided by the program or received by the recipients, such as the distribution of ice cleats, educational workshops, etc. If the program is implemented successfully, its outputs are then expected to affect various *outcomes*, as detailed in the latter part of the model. Assuming the distribution of ice cleats leads to an increase in ice cleat use, it should be able to reduce fall-related injuries. It may also lead to increased walking, thereby promoting better health and well-being among the senior citizens [6].

The top half of the model describes conditions that we classified as necessary conditions for an ice cleat program to be able to have some effect on health and well-being. The bottom half describes hypothesized modifying conditions that may affect the probability of a successful implementation and/or increase the size of the impact on health outcomes.

## Analysis of process theory

In the following text, we discuss the model's components in detail with reference to the empirical results presented above and relevant literature identified in our search, aiming to discuss the plausibility of the proposed logic model and identify factors that can influence successful implementation and/or modify the impact of ice cleat programs.

**Inputs.** A formal decision to implement the program is required at the municipal level, as governed by the Swedish Local Government Act [18]. According to our survey, 57 municipalities had considered implementing an ice cleat distribution program but decided not to do so. The most common reasons provided for not introducing programs was that such programs were not considered to be within the scope of the municipalities responsibilities (n = 19), that economic ventures for the municipality were used elsewhere (n = 13), that the municipality did not have the resources required to introduce such a program (n = 9), or that a motion was put forth but rejected politically (n = 9). There were also municipalities that pointed out there is a lack of evidence for the effectiveness of ice cleat programs (n = 7). Some municipalities also referred to a paragraph in the Local Government Act that states that municipalities and regions must treat all citizens equally unless there are objective reasons to provide special services to certain groups (n = 6). Thus, many potential programs may fail even before they are started if these issues are not properly navigated.

A municipality that decides to implement an ice cleat program will also require a sufficient amount of ice cleats to distribute to their target population and other resources [8,9] (e.g., allocation of time to staff to perform the program activities, sufficient funding to complete the activities, appropriate knowledge, devise communication and distribution plans, etc.). One may assume that such resources are appropriately allocated upon a formal decision to implement the program. However, external factors may hamper implementation. For instance, one municipality reported that their ice cleat order was stuck in customs for several weeks (U. Stefansson, personal communication, August 21, 2019). Such delays may affect the program's success, given that the timing of the distribution efforts may be crucial to ensure a sufficient reach before the beginning of the winter season.

**Activities.** Assuming the required inputs are sufficiently secured (Fig 1), the program staff must implement the distribution plan, including when, how, and where to distribute the ice cleats to the population. According to our survey, most intervention municipalities distributed ice cleats at municipal-owned buildings (such as libraries or city halls; Table 1). Others relied on partnerships with retailers such as stores, pharmacies, and other businesses that provide ice cleats. The rest relied on partnerships with pensioner organizations or distributed ice cleats at

other locations (e.g., during an annual "falls prevention week"). As noted above, it appears that municipalities with above-average reach tended to partner with, for instance, retailers and health care centers.

Another necessary condition for a successful distribution is that the target population is aware of the program's existence [8,9,19] and that they are eligible for a free pair of ice cleats. A majority of the intervention municipalities communicated the program using multichannel campaigns [19] like traditional media, social media, mail, personal communication, etc., to create knowledge and spread awareness about the program. Proper communication about how and when to use ice cleats may encourage more individuals to use them (e.g., how to attach them to regular footwear, disassemble when entering indoors, etc., which can reduce perceived barriers to use [20] and improve self-efficacy [21]). If the communication about the program is supplemented with convincing messaging about the problem (e.g., lack of mobility, risk of falling) and how ice cleats may help combat these issues, communication theory and the Health Belief Model would suggest that the program may persuade more individuals about the utility of ice cleats [19,20].

**Outputs.**   Assuming the activities are performed successfully, the expected output is that the ice cleats are distributed to a meaningful share of the target population. As noted above, we found that the average intervention municipality distributed ice cleats to about 40% of the eligible population residing in the municipality. One factor that appears to increase reach is the availability of many distribution sites around the municipalities combined with a communications plan covering the entire target population. For instance, Gothenburg reached approximately 60% of its older citizens by mailing coupons to all eligible citizens and spreading the pick-up locations throughout the city [7]. On the other hand, one municipality limited its distribution in a one-week falls prevention festival and reached as little as 1 percent of the target population. Overall, our survey data suggest that the main predictor of reach is the number of purchased ice cleats per population size (9 in 10 ice cleat pairs will eventually be collected; Table 1). While factors related to communication and distribution may play an essential role in increasing the program's impact, it appears that the main barrier to widespread distribution is a lack of scale; shortage of resources, and/or the lack of ambition to reach the entire population.

## Analysis of impact theory

**Effects on initial behavior change.**   At the population level, the intervention impact is a product of the effect on the ice cleat use and the causal effect of ice cleat use on relevant health outcomes (e.g., injury rates, increased walking). Thus, a key component that will affect the magnitude of the intervention effect on all outcomes is the increase in *new* ice cleat users. This increase is not necessarily equivalent to the number of people who obtain a free pair of ice cleats.

The Health Belief Model suggests that individuals who identify themselves as fall-prone are assumed to be more likely to search for risk-reducing measures to prevent a fall, suggesting that fall-prone individuals may be more easily convinced by the benefits of the intervention and thus act in line with the program [20]. Another key motivational driver for the use of safety equipment is the subjective perception of its efficacy [22]. However, individuals who share these perceptions may also be more likely to use ice cleats without a distribution program. Ceiling effects are also a potential concern, as a program cannot possibly have an effect if everyone in the population already uses ice cleats. According to a national survey from 2014, about 40% of the Swedish population above the age of 65 report seldom or never using ice cleats [23], which implies that there are still individuals who may benefit from a

program. However, usage rates tend to be higher in municipalities with more snow days per year (closer to 75% self-reported use) [23], which may limit the potential impact in high-risk municipalities.

In addition to these conceptual uncertainties, there is also no direct evidence to support the claim that a meaningful proportion of current non-users will start using ice cleats as a consequence of a distribution program. Indirect evidence can, however, be found in a quasi-experimental evaluation from Gothenburg, where the rate of pedestrian fall injuries among older adults appears to have decreased as an effect of the program [7]. Still, several of our interviewees noted that ice cleats could be challenging to apply and pointed out that decision-makers should focus on distributing user-friendly devices. Otherwise, there is a risk that the collected devices will be left unused.

**Maintained behavior change.** Another key aspect of behavior change is its maintenance (i.e., longevity). Upholding a behavior change in the long term after an intervention appears to be difficult in general [24,25], and the behavior change associated with ice cleat programs may therefore be short-lived [19]. Our interview data support this notion, especially if the ice cleats are of poor quality or difficult to apply. On the other hand, our interviewees also noted that ice cleats could enable an active and safer lifestyle during the winter, which points to a potentially strong motivational driver for continued use. A free-text response to our survey from one municipality also showed that recipients came back asking for a new pair of devices in exchange for handing in the old broken ones. The demand for ice cleats led to the municipality distributing more devices than there were citizens in the eligible age group (approximately 1.1 pairs per older citizen). In addition, evidence from Gothenburg suggests an effect on injury rates may be limited to the first implementation year [7]. Together, these pieces of evidence indicate that individuals may change their behavior initially and then return to their normal behavior or that the ice cleats tend to break after a while and need to be replaced.

The perceived quality of the distributed ice cleats and their user-friendliness may play an important role in maintaining any initial changes in ice cleat use. Previous research suggests that ease of use and perceived utility may influence the perception of ice cleats [26,27]. A good initial experience is also likely to increase the chances of prolonged use [24], which may lead to a long-term behavior change in subsequent winters. It was also stressed by our interviewees that no matter what type of device, the ice cleats must be user-friendly and easy to apply. According to our survey, most municipalities distributed whole foot devices (72.9%; Table 1), which appears appropriate given evidence from previous preference studies that suggest that whole foot and heel devices are the preferred types of ice cleats [27–29]. Despite this apparent alignment between program designs and preference data, our interviewees advised caution against this type of design as it deprives individuals of the opportunity to freely choose devices of their choice, which can hinder new users from collecting and using the ice cleats. However, they noted that they would recommend them to friends if they found an appropriate device. These results are consistent with a previous survey of ice cleat users in northern Sweden [30]. Behavior change theory also suggests individuals are more likely to follow a recommended behavior if it is suggested by trusted peers [21]. Thus, providing high-quality and user-friendly ice cleats may increase the magnitude of a program's impact on ice cleat use. It may also have other spin-offs, such as increased reach and use via communication between peers.

**Effects on health outcomes.** A key hypothesized impact of the ice cleat programs is a reduction in the risks of ice and snow-related fall injuries. There is a strong theoretical argument in support of a causal effect on fall risk as long as ice cleats are worn, given that increased friction decreases slipperiness [27,31,32], which is the key mechanism behind ice-related falls. We also identified two empirical studies supporting this claim. One was a randomized trial on a northern US sample of older, fall-prone adults, which indicated approximately a 50%

decrease in fall risk and a 90% decrease in risk of injurious falls. The other was an observational study from a general adult sample in northern Sweden, which showed a 37% lower fall risk per kilometer among ice cleat users. Overall, this evidence supports the assumption that the use of ice cleats decreases the risk of fall-related injuries during winter conditions [29,30].

Another potential effect is an increase in walking during the winter months. Our interviewees expressed an understanding of the positive health effects of upholding an active lifestyle. If the ice cleat programs increase walking, they may also affect general health and well-being [33,34]. However, the empirical evidence of the effects of ice cleats on walking is not as strong as the evidence for an effect on injury risks. We found one study that indicates that ice cleat users walk more, but it is unclear whether ice cleats lead to additional walking or if active people are more interested in using ice cleats [30]. It is worth noting that an increase in walking means an increase in risk exposure, which may offset the number of injuries prevented at the population level. However, the randomized trial reported in reference [29] used diary-days, not distance walked, as their exposure denominator and still showed a considerable impact on injury rates, which implies that the potential impact on walking does not offset the effect on injuries completely. Overall, our analysis of the evidence on health effects suggests that ice cleats are likely to be beneficial if worn.

### Summary of identified barriers to program success

To conclude our analysis, we summarize the main barriers to implementation and successful impact in Fig 2. The barriers highlighted in the figure are those that we interpret as either necessary for widespread reach as identified in our analyses or critical assumptions of the logic model that lack sufficient empirical support to be taken for granted.

## Discussion

This is the first broad-scale process evaluation of Swedish municipal ice cleat distribution programs, an intervention that aims to decrease the risk of pedestrian falls among older adults and promote active transportation during winter conditions. We triangulated data from several

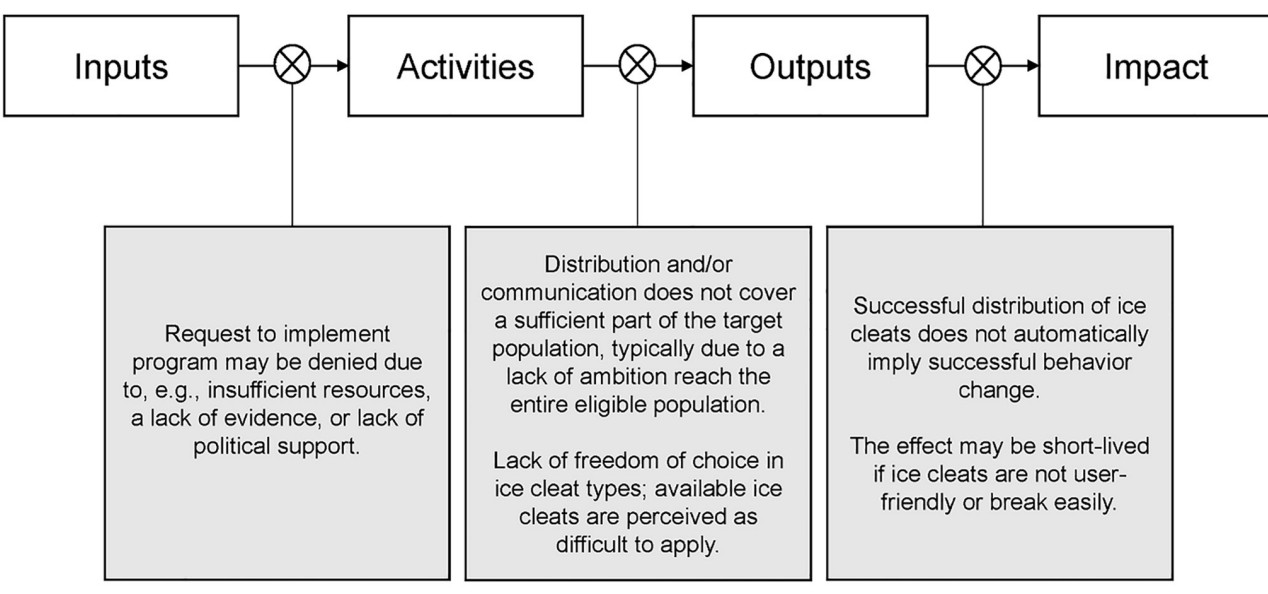

**Fig 2. Summary of critical barriers to program success identified in the data and program theory analysis.**

sources to describe the characteristics of Swedish municipal ice cleat programs and assess their implementation. Our program theory analysis also contributed to the identification of potential effect modifying factors that may help improve the design of future programs. Our study implies that it is important to secure sufficient political support and provide more robust evidence on the effectiveness of ice cleat programs and that the programs should aim to cover the entire population to ensure that population-level impact is possible. The programs should carefully consider the user-friendliness of the offered ice cleats (e.g., ease of application); otherwise, the effect on health outcomes–if any–is likely to be short-lived (Fig 2).

## Strengths and limitations

Our broad focus enabled the use of data from all existing programs in Sweden. However, the scope of the study also has its downsides. For instance, in-depth case studies may help shed additional light on implementation issues and their potential solutions. Our general interpretation is that the implementation of these programs is relatively straightforward, as is evidenced by their reach, and that the main issues may lie elsewhere (e.g., in actually achieving and maintaining a behavior change in the population). This study's retrospective design means that we could not collect data on ice cleat use before and after the interventions to investigate effects on behavior. Even though 90 percent of the purchased ice cleats were distributed according to our data, there is currently no direct evidence that the target population actually uses the ice cleats after distribution. Not only was it stressed by our interviewees, but the effects on behavior change may be short-lived [7,19], particularly if the distributed ice cleats are not perceived as user-friendly (see the Focus-group interviews section for details). Also, prolonged maintenance of behavior change can be challenging to uphold [24]. While evidence of a short-run impact on ice-related fall injuries has been observed in one municipality [7], our logic model highlights several factors related to implementation and context that may influence the size and longevity of the impact. Future research should therefore focus on a broader impact evaluation of existing programs in order to assess their average impact on behaviors and health outcomes, as well as factors that influence effect size. Our study can provide important input for such analyses.

## Conclusion

The implementation of the majority of existing ice cleat programs in Sweden appears to have been successful in terms of reach. However, significant uncertainty remains as to whether they have had a meaningful impact on ice cleat use, highlighting the importance of a comprehensive evaluation of the impacts of these programs.

## Supporting information

**S1 Appendix. Semi-structured interview guide.**
(DOCX)

**S1 Table. Key components of the proposed logic model for ice cleat distribution programs, with theoretical arguments, counterarguments and sources of evidence.**
(DOCX)

## Acknowledgments

We are grateful to study participants for taking their time to provide valuable input to this study and the Swedish municipalities that participated in the survey. We also would like to

thank Finn Nilson and Mathilde de Goër de Herve for their valuable feedback on an earlier draft of this paper.

## Author Contributions

**Conceptualization:** Robin Holmberg, Johanna Gustavsson, Carl Bonander.

**Data curation:** Robin Holmberg, Johanna Gustavsson, Carl Bonander.

**Formal analysis:** Robin Holmberg, Johanna Gustavsson, Carl Bonander.

**Funding acquisition:** Johanna Gustavsson, Carl Bonander.

**Investigation:** Robin Holmberg, Johanna Gustavsson.

**Methodology:** Robin Holmberg, Johanna Gustavsson, Carl Bonander.

**Resources:** Robin Holmberg, Johanna Gustavsson, Carl Bonander.

**Software:** Robin Holmberg, Carl Bonander.

**Supervision:** Johanna Gustavsson, Carl Bonander.

**Writing – original draft:** Robin Holmberg.

**Writing – review & editing:** Johanna Gustavsson, Carl Bonander.

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
