## [Decision Letter · Decision Letter 0]

27 Apr 2021

PONE-D-21-09111

Evaluation of the design and implementation of municipal ice cleat distribution programs for the prevention of ice-related fall injuries among older adults in Sweden

PLOS ONE

Dear Dr. Holmberg,

Thank you for submitting your manuscript to PLOS ONE. After careful consideration, we feel that it has merit but does not fully meet PLOS ONE’s publication criteria as it currently stands. Therefore, we invite you to submit a revised version of the manuscript that addresses the points raised during the review process.

Please consider all comments.

We look forward to receiving your revised manuscript.

Kind regards,

Ahmed Mancy Mosa, Ph.D.

Academic Editor

PLOS ONE

Journal Requirements:

2. When reporting the results of qualitative research, we suggest consulting the COREQ guidelines: http://intqhc.oxfordjournals.org/content/19/6/349

In this case, please consider including more information on the number of interviewers, their training and characteristics; and please provide and clarify whether an interview guide used.

Furthermore, in the Methods section, please include details regarding the date range of the literature search conducted.

Reviewers' comments:

Reviewer's Responses to Questions

**Comments to the Author**

1. Is the manuscript technically sound, and do the data support the conclusions?

Reviewer #1: Yes

Reviewer #2: No

2. Has the statistical analysis been performed appropriately and rigorously? 

Reviewer #1: Yes

Reviewer #2: Yes

3. Have the authors made all data underlying the findings in their manuscript fully available?

Reviewer #1: Yes

Reviewer #2: Yes

4. Is the manuscript presented in an intelligible fashion and written in standard English?

Reviewer #1: Yes

Reviewer #2: Yes

5. Review Comments to the Author

Reviewer #1: I recommend this manuscript , the ice cleat program showed a good results in protecting people specially the elderly in decreasing the risk of road injury during winter time.

And the Swedish municipal ice cleat distribution programs is targeting mostly the elderly people over 75 years to support them with ice cleat that will lead to more protection and lowering the risk of injury.

Reviewer #2: Clearly a great deal of effort went into this study, however a number of areas need clarifications:

- Were there set targets at the onset of the intervention. Measurements of the success will be best adjudged by how well these targets were achieved or otherwise e.g. did they target all potential users or did they seek to meet a certain percentage? The quoted figures in this report carry little weight in themselves if not in the context of these targets.

- How many people used or had access to the cleats before the intervention?

- What were the fall rates before and after the intervention?

6. PLOS authors have the option to publish the peer review history of their article (what does this mean?). If published, this will include your full peer review and any attached files.

Reviewer #1: **Yes: **Dr. Liqaa Raffee

Reviewer #2: No

---

## [Author Response · Author response to Decision Letter 0]

17 May 2021

Editor: Thank you for your valuable feedback. We have incorporated your input into the revised manuscript. For further information, we refer you to the "Response to Reviewers" document.

Reviewer#1: Thank you for your taking the time to review our manuscript and for your positive feedback.

Reviewer#2: Thank you for highlighting that the manuscript was in need of clarification. We have incorporated changes in the manuscript.

---

## [Editor Report · Decision Letter 1]

28 May 2021

Evaluation of the design and implementation of municipal ice cleat distribution programs for the prevention of ice-related fall injuries among older adults in Sweden

PONE-D-21-09111R1

Dear Dr. Holmberg,

We’re pleased to inform you that your manuscript has been judged scientifically suitable for publication and will be formally accepted for publication once it meets all outstanding technical requirements.

Kind regards,

Ahmed Mancy Mosa, Ph.D.

Academic Editor

PLOS ONE
---

## [Editor Report · Acceptance letter]

15 Jun 2021

PONE-D-21-09111R1 

Evaluation of the design and implementation of municipal ice cleat distribution programs for the prevention of ice-related fall injuries among older adults in Sweden 

Dear Dr. Holmberg:

I'm pleased to inform you that your manuscript has been deemed suitable for publication in PLOS ONE. Congratulations! Your manuscript is now with our production department. 

Kind regards, 

on behalf of

Dr. Ahmed Mancy Mosa 

Academic Editor

PLOS ONE